



# A 15 million-year long record of phenotypic evolution in the heavily calcified coccolithophore *Helicosphaera* and its biogeochemical implications

Luka Šupraha[1, a], Jorijntje Henderiks[1, 2]

[1] Department of Earth Sciences, Uppsala University, Villavägen 16, SE-752 36 Uppsala, Sweden.
[2] Centre for Ecological and Evolutionary Synthesis (CEES), Department of Biosciences, University of Oslo, P.O. Box 1066 Blindern, 0316 Oslo, Norway.
[a] Present address: Section for Aquatic Biology and Toxicology (AQUA), Department of Biosciences, University of Oslo, P.O.
Box 1066 Blindern, 0316 Oslo, Norway

*Correspondence to*: Luka Šupraha (luka.supraha@ibv.uio.no)

**Abstract.** The biogeochemical performance of coccolithophores is defined by their overall abundance in the oceans, but also by a wide range in cell size, degree of calcification and carbon production rates between different species. Species' "sensitivity" to environmental forcing has been suggested to relate to their cellular PIC:POC ratio and other physiological constraints.

Understanding both the short and longer-term adaptive strategies of different coccolithophore lineages, and how these in turn shape the biogeochemical role of the group, is therefore crucial for modeling the ongoing changes in the global carbon cycle. Here we present data on the phenotypic evolution of a large and heavily-calcified genus *Helicosphaera* (order Zygodiscales) over the past 15 million years (Ma), at two deep-sea drill sites from the tropical Indian Ocean and temperate South Atlantic. The modern species *Helicosphaera carteri*, which displays eco-physiological adaptations in modern strains, was used to

benchmark the use of its coccolith morphology as a physiological proxy in the fossil record.

Our results show that, on the single-genotype level, coccolith morphology has no correlation with physiological traits in *H. carteri*. However, significant correlations of coccolith morphometric parameters with cell size and physiological rates do emerge once multiple genotypes or closely related lineages are pooled together. Using this insight, we interpret the phenotypic evolution in *Helicosphaera* as a global, resource limitation-driven selection for smaller cells, which appears to be a common

adaptive trait among different coccolithophore lineages, from the warm and high-$CO_2$ world of the middle Miocene to the cooler and low-$CO_2$ conditions of the Pleistocene. However, despite a significant decrease in mean size, *Helicosphaera* kept relatively stable PIC:POC (as inferred from the coccolith "aspect ratio") and thus highly conservative biogeochemical output on the cellular level. We argue that this supports its status as an "obligate calcifier", like other large and heavily-calcified genera such as *Calcidiscus* and *Coccolithus*, and that other adaptive strategies, beyond size-adaptation, must support the

persistent, albeit less abundant, occurrence of these taxa. This is in stark contrast with the ancestral lineage of *Emiliania* and *Gephyrocapsa*, which not only decreased in mean size but also displayed much higher phenotypic plasticity in degree of calcification while becoming globally more dominant in plankton communities.



## 1 Introduction

Coccolithophores (calcifying Haptophyte algae) are a globally abundant group of marine phytoplankton and an important component of the biogeochemical carbon cycle. Their calcification (Monteiro et al., 2016), primary production (Rousseaux and Gregg, 2013) and deep-sea burial of calcite scales (coccoliths) (Baumann et al., 2004; Ziveri et al., 2007) have been crucial for the evolution of past and modern ocean chemistry, and are still alleviating the negative effects of rising atmospheric and oceanic $CO_2$ levels (Ridgwell and Zeebe, 2005). Their net biogeochemical impact, which is commonly summarized as a

cellular balance of inorganic (PIC) and organic (POC) carbon production rates (*i.e.* PIC:POC ratio), is highly susceptible to environmental forcing such as high-temperature stress (Gerecht et al., 2017, 2014; Rosas-Navarro et al., 2016), nutrient limitation (Bolton and Stoll, 2013; Gerecht et al., 2015; Müller et al., 2017) and ocean acidification (Ridgwell et al., 2009; Riebesell et al., 2000). Understanding how coccolithophores adapt to environmental changes and how adaptive strategies shape their biogeochemical impact is thus essential for modeling the carbon cycle dynamics in the projected warmer, high-$CO_2$

oceans of the near future (Bopp et al., 2013; Doney et al., 2004; Feely et al., 2004).

Deep-sea sediments hold a ~200 million year (Ma) long fossil record of coccoliths (Bown et al., 2004). Some morphometric features of coccoliths, such as length and thickness, correlate with biogeochemically relevant traits: cell size (Henderiks, 2008), growth rates (Gibbs et al., 2013) and the production rates of organic (POC) and inorganic (PIC) carbon (Bolton et al., 2016; McClelland et al., 2016). Combined with morphospecies-diversity and community composition that is readily inferred from

the fossil record (Bown et al., 2004; Knappertsbusch, 2000; Suchéras-Marx and Henderiks, 2014), coccolith morphometry could therefore help elucidate evolutionary processes from the cellular- to the community-level. In this context, the transition from the warm and high-$CO_2$ world of the middle Miocene (~15 Ma) to the colder, low-$CO_2$ conditions of the Pleistocene (Super et al., 2018; Zachos et al., 2001) provides an optimal case-study for investigating the long-term evolutionary adaptation of modern coccolithophore lineages to relevant climate-forcing, and understanding the biogeochemical implications of their

evolutionary adaptation. This was a period of large-scale shifts in coccolithophore community structure, abundance and biogeochemical impact (Hannisdal et al., 2012; Si and Rosenthal, 2019), as also evidenced by a macroevolutionary decrease in cell size (Suchéras-Marx and Henderiks, 2014) and cellular calcification rates (Bolton et al., 2016).

To date, studies of coccolithophore evolutionary adaptation have mostly focused on members of the Noelaerhabdaceae family, which dominated coccolithophore communities during the middle Miocene and Pleistocene (Henderiks and Pagani, 2007;

Suchéras-Marx and Henderiks, 2014). While their most important modern representative *Emiliania huxleyi* is highly abundant in today's oceans, this lineage represents only a small fraction of the morphological, ecological and physiological diversity of coccolithophores (Young et al., 2005). The Noelaerhabdaceae also have a unique evolutionary history with an early Jurassic divergence (~195 Ma) from the other calcifying haptophytes (Medlin et al., 2008). Clearly, there is a need to investigate evolutionary adaptation across the range of diversity and include other prominent coccolithophore lineages to determine

whether the evolutionary patterns observed in the Noelaerhabdaceae are universal for the group as a whole.



This study focuses on the phenotypic evolution of the heavily calcified coccolithophore genus *Helicosphaera* (order Zygodiscales; Fig. 1). This genus was a constant component of coccolithophore communities during the Cenozoic (Aubry, 1988) and is still today an important contributor to oceanic carbonate fluxes (Baumann et al., 2004; Menschel et al., 2016; Ziveri et al., 2007). The most common modern species within this genus, *Helicosphaera carteri* (Wallich 1877) Kamptner, 1954, can be grown in culture (Sheward et al., 2017; Šupraha et al., 2015) and has a long fossil record spanning over 20 Ma (Aubry, 1988). Earlier experimental studies have found a range of strain-specific physiological traits in *H. carteri*, likely related to local eco-physiological adaptation (Šupraha et al., 2015). In addition, this species is considered highly susceptible to ocean acidification due to its heavily calcified coccoliths and high PIC:POC ratio (Gafar et al., 2019). Finally, both modern and fossil *Helicosphaera* morphospecies are well-defined and identifiable under a light microscope, which allows for long-term tracking of speciation events, community dynamics and phenotypic evolution at the morphospecies-level. The main aims of this study are to (I) benchmark the use of coccolith morphology as a potential physiological proxy in *Helicosphaera* based on the coccolith morphology of physiologically distinct *H. carteri* strains in culture, (II) explore the adaptive strategies of the genus and main evolutionary drivers during the past 15 Ma, and (III) compare the evolutionary patterns of *Helicosphaera* with other major coccolithophore lineages and discuss the biogeochemical implications of their evolutionary adaptation.

## 2 Material and Methods

### 2.1 *Helicosphaera carteri* cultures

Culture data, obtained from batch experiments presented in Šupraha et al. (2015), are herein used as a bench-mark for phenotypic plasticity in modern *H. carteri*. Two strains were isolated from contrasting environments, the eutrophic South Atlantic Ocean (strain RCC1323) and oligotrophic western Mediterranean Sea (strain RCC1334) (Fig. 2a). These strains exhibited different physiological traits such as cell size, growth and calcification rates (PIC production rates). Strains were acclimated to experimental conditions for 10 generations prior to inoculation. Experiments were run in triplicate at a temperature of 17 °C and an irradiance of ~160 µ mol m$^{-2}$ s$^{-1}$ under a 14:10 h light:dark cycle. Further details of the experimental setup are given in Šupraha et al. (2015). All phenotypic and physiological data used in the present study were obtained from the exponentially growing, non-limited control experiments only.

### 2.2 Deep-sea sediment records

Fossil time series data were collected from pelagic foraminiferal nannofossil oozes and chalks recovered at two deep-sea drill sites (Fig. 2a). Deep Sea Drilling Project (DSDP) Leg 74 Site 525 is located in the southeastern Atlantic Ocean, at the Walvis Ridge (29°04.24' S, 02°59.12' E; water depth 2467 m) (Moore et al., 1984). Ocean Drilling Program (ODP) Leg 115 Site 707 is located in the tropical western Indian Ocean, at the Mascarene Plateau (07°32.72' S, 59°01.01' E; water depth 1552 m) (Backman et al., 1988). The sites were selected to represent contrasting productivity levels and climatic conditions. Modern-day primary production estimates are on average 50-100 g C m$^{-2}$ yr$^{-1}$ higher in the South Atlantic than in the equatorial Indian



Ocean (Antoine et al., 1996; Beaufort et al., 1997; Fischer et al., 2000). Consistently higher carbonate mass accumulation rates at Site 525 compared to Site 707 (Suchéras-Marx and Henderiks, 2014) confirm consistent site-to-site carbonate production contrasts during the geological past. Regional differences in mean sea surface temperature extend back at least 10 million years

(Fig. 2b). Age models are based on calcareous nannoplankton biostratigraphy, updated to the most recent geological time scale (Gradstein et al., 2012) (Table S1). Linear sedimentation rates were estimated between age-depth tie-points.

### 2.3 Coccolith morphometry

*Helicosphaera carteri* coccoliths harvested from triplicate cultures were filtered on cellulose nitrate filters (0.8 μm, Whatman) after dispersing the coccoliths with a Triton–NaOCl treatment (Paasche et al., 1996). Filters were dried at 60°C and mounted

with Canada Balsam (Merck, USA) on microscope slides, rendering transparent filters under polarized light. Microscope slides for fossil coccolith morphometry were prepared from fine fraction (<38μm) sediment using the spraying technique (Henderiks and Törner, 2006) for a total of 36 (Site 525) and 38 (Site 707) stratigraphic levels. Sub-samples of freeze-dried bulk sediments were wet-sieved with buffered distilled water into three separate size fractions (<38 μm, 38–63 μm and >63 μm). The fine fraction, dominated by coccoliths, was collected in suspension, concentrated on cellulose acetate membrane filters using a

Millipore vacuum pump and oven-dried at 50°C.

In total, 331 culture-harvested and 4115 fossil *Helicosphaera* specimens were analysed using a Leica DM6000B microscope equipped with polarizing filters placed at different angles (Beaufort et al., 2014). Besides size measurements, mean grey level (MGL) was used as a proxy for a change in coccolith thickness based on the birefringence of calcite (Beaufort, 2005). Standard imaging and light settings were ensured during the course of image collection (Intensity=200, Field=10, Aperture=10). Image

collection was performed within a short time frame in order to minimize the effect of light bulb aging. For each sample, at least 50 *Helicosphaera* coccoliths were imaged at 1000x magnification with a SPOT Flex color digital camera (Diagnostic Instruments Inc.; μm-scale calibrated pixel resolution 0.061μm [1.2x C-mount] or 0.074μm [1.0x C-mount]) under three different orientations of the polarizer (0°, 35°, and 45°). Coccoliths were randomly selected and identified to morphospecies level. Images taken with the three orientations of the polarizer were merged in a single 8-bit greyscale image, showing no

extinction cross (Beaufort et al., 2014) (Fig. 1). Images from each sample were analyzed as a batch using a custom-made macro in ImageJ software v1.47c. It determined the MGL of the selected area, measured in grey-level values of 0 (black) to 255 (white), and performed size measurements on individual coccoliths (area, perimeter and length and width of the fitted ellipse). The calculation of coccolith thickness and volume was conducted following the protocol described in Beaufort et al. (2014):

Thickness [μm] = MGL * (1.55/255) [μm]                                                           (Eq. 1)

Volume [μm$^3$] = Thickness [μm] * Area [μm$^2$]                                                (Eq. 2)

It should be noted that all estimates of *Helicosphaera* coccolith volume are systematically underestimated, but that this does not preclude accurate, relative comparisons between samples. Variations in area estimates are reliable within the constraints



of image segmentation, but *Helicosphaera* coccoliths (also known as "helicoliths") are not entirely birefringent under crossed or circular polarized light due to the crystallographic orientation of calcite elements; the proximal plate and the "blanket" that

covers most of the central area are birefringent in analysis, but the "flange" that extends from the proximal plate and underlies the blanket is composed of so-called V-units that are only partially birefringent in the same orientation on a microscope slide (Fig. 1) (Beaufort et al., 2014; Young et al., 2004). In addition, parts of the central area exceed 1.55µm (or 255 (white) value; Beaufort et al., 2014), showing faint yellow birefringence colors that render grey values (<255) in 8-bit, thus lowering the mean grey-value estimate calculated from all pixels in one specimen). An examination of random specimens from both sites

performed in ImageJ using the "Threshold color" function has shown that the yellow signal from the thickest parts of the central area usually accounts for up to 5% of surface area in medium-sized species (e.g. *H. carteri* and *H. sellii*) and up to 10%-15% in the largest species *H. granulata*. Therefore, *Helicosphaera* volume estimates are not converted to calcite mass (multiplied by the density of calcite, 2.7 pg/µm$^3$). Instead, in analogy with previous studies on *Coccolithus pelagicus* (Cubillos et al., 2012; Gerecht et al., 2014), we report on changes in "coccolith volume index" (CVI, in ln µm$^3$) that scales with other

coccolith size and shape parameters. Statistically significant changes in CVI are meaningful indicators, if not a muted expression, of the true phenotypic changes in both modern and fossil *Helicosphaera* assemblages.

Finally, we compiled morphometric time series data for *Calcidiscus* (Knappertsbusch, 2000) and *Reticulofenestra* (Bolton et al., 2016; Hannisdal et al., 2012) from tropical and mid-latitude locations to compare patterns of phenotypic variation and evolution across major coccolithophore clades. This comparison includes a total of 4907 *Reticulofenestra* coccoliths measured

in 37 samples at Site 525. The latter database is limited to linear size measurements (length) only, because it was collected with a different polarized light microscopy and digital image analysis setup (Henderiks, 2008; Henderiks and Törner, 2006).

**2.4 PIC:POC and geometric proxies for cell physiology**

Cellular production rates of particulate inorganic (PIC) and organic carbon (POC) were determined for *H. carteri* in the exponentially growing control experiments as a product of growth rate and the corresponding cellular elemental quotas

(Šupraha et al., 2015). The molar ratio PIC:POC is widely used to assess the relative balance between calcification and photosynthesis, which in turn informs about the net $CO_2$ uptake or release by coccolithophores (review by Ridgwell et al., 2009) and references therein).

These physiological rates cannot be directly measured in the fossil record. Nevertheless, within a theoretical framework of the relationships between cell geometry, coccolith dimensions and coccolith coverage, cellular levels of organic and inorganic

carbon (calcite) quota can still be estimated with a range of approaches (Bolton et al., 2016; Gibbs et al., 2013, 2018; Henderiks, 2008; Jin et al., 2018; McClelland et al., 2016).





In this study, we use the approach described by McClelland et al. (2016) who correlated a coccolith shape index, the "aspect ratio" (AR), with PIC:POC ratios measured in modern strains of *Emiliania huxleyi* and *Gephyrocapsa oceanica*, deriving the following power-law relationship:

$PIC:POC = e^{3.5\pm0.2} AR^{1.12}$ (Eq. 3)

where

$AR = Thickness / \sqrt{(Area)}$ (Eq. 4)

Despite the fact that coccolith thickness in *Helicosphaera* is systematically underestimated, we took a similar approach to investigate relative changes in shape (AR) and cellular degree of calcification (PIC:POC) between mean phenotypes.

**2.5 Coccolith abundance counts and paleo-fluxes**

For the quantitative analysis of fossil coccolith assemblages, microscope slides were prepared directly from freeze-dried bulk sediments using the drop technique (Bordiga et al., 2015). The 74 samples were analysed at 1000x magnification under an Olympus BX53 polarized-light microscope. A minimum of 300 coccoliths was counted and at least 20 fields of view (FOV) were analysed in each sample to determine the relative contribution of the major coccolithophore taxa. The absolute abundance

of coccoliths (in N/g) was calculated using the formula:

$AA_{coccolith} = (N \times A)/ (f \times n \times W)$ (Eq. 5)

Where N is the number of coccoliths counted, A is the area of the coverslip, f is the area of one FOV ($0,0367$ mm$^2$), n is the number of FOVs counted and W is the weight of the sediment on the coverslip (g).

Fluxes of *Helicosphaera* spp. and other taxa (in N m$^{-2}$ yr$^{-1}$) were calculated from the absolute abundances and the local mass

accumulation rates (MAR; data from Suchéras-Marx and Henderiks (2014) with a reproducibility of $\pm15\%$:

$Flux_{coccolith} = AA_{coccolith} (N/g) \times MAR (g m^{-2} yr^{-1})$ (Eq. 6)

Past primary productivity estimates (PP, in g C m$^{-2}$ yr$^{-1}$) were calculated from the relative abundance of the lower photic zone species *Florisphaera profunda* (in percent, %*Fp*), taking the 95% confidence intervals as upper and lower inputs, *cf.* (Beaufort et al., 1997):

$PP = 617 – [279 \times log (\%Fp + 3)]$

(Eq. 7)



## 2.6 Statistical analysis

Raw biometry data (μm-scale) were transformed to their natural logarithm (ln μm-scale) before calculating sample means and variances, when investigating allometric relationships and comparing temporal trends within the time series.

One-tailed t-tests were performed in R version 3.2.2. using the "t.test" command from the *Stats* package. Paired t-tests were used to compare means between culture strains and between the fossil measurements. For the latter, site-to-site comparisons were made at the genus and (morpho)species level. P-values < 0.05 were considered statistically significant. The 95%

confidence intervals for relative abundance estimates were calculated in PAST version 3.20 freeware (Hammer et al., 2001; Suchéras-Marx et al., 2019).

## 3. Results

### 3.1 *Helicosphaera carteri* allometry

The scaling between coccolith size and thickness estimates of *H. carteri* shows slightly negative allometry and is highly similar

in modern strains and fossil specimens (Fig. 3a,b). Mean phenotype dimensions in modern single genotypes fall within the lower range of the fossil measurements. Differences in mean coccolith traits between the South Atlantic and Mediterranean strains are not statistically significant, or only borderline significant (paired t-tests, *p*-values= 0.029-0.039; Table S2). The range of shapes (AR) and dimensions (CVI) in cultured *H. carteri* largely overlap with its fossil counterparts (Fig. 3c,d). The Mediterranean strain (RCC1334), however, produced higher proportions of small and lightly calcified coccoliths ('outliers'

from the distribution; Fig. 3d). Likewise, the raw fossil data ranges indicate that the Indian Ocean Site 707 hosted higher proportionas of small *H. carteri* specimens than South Atlantic Site 525 (Fig. S1).

### 3.2 Mean phenotypic evolution of the *Helicosphaera* genus

The fossil time series data, now considering all measurements of *H.carteri* and other fossil *Helicosphaera* morphospecies, show long-term trends in mean CVI and AR at both sites (Fig. 4a,b). Linear regression of sample means vs. age suggests a

16% decrease in mean size and CVI (ln-scale) from the middle Miocene to late Pleistocene at South Atlantic Site 525 (r=0.849), which corresponds to a 46% decrease in mean volume on the linear scale ($\mu m^3$). The same trend is less pronounced at the tropical Indian Ocean Site 707, with a 9% decrease in mean size and CVI (27% decrease in volume) over the same time interval (r=0.525). Mean shape (AR) estimates shows an opposite trend with age at both sites, implying a 24% (525; r=0.946) and 18% (707; r=0.815) increase in aspect ratio over the past 15 million years. The AR is mainly driven by changes in size, since the

decrease in thickness over time is less pronounced (Fig. S2) suggesting that the *relative* degree of calcification increased as coccolith dimensions (and thus mass per individual coccolith) decreased at both sites.



When comparing all fossil measurements over the past 15 Ma, mean CVI was significantly smaller at Site 707 than at Site 525 (paired t-test; t=20.36, df=3863.8, *p*-value <0.001). The Indian Ocean record also reveals larger variation in Miocene-Pliocene sample means compared to the more gradual trend observed in the South Atlantic, so that site-to-site offsets in mean CVI varied over time (Fig. 4a). These offsets were largest throughout the middle Miocene, after which mean CVI reached more similar values that persisted during the late Miocene between ~8.5 – 5.5 Ma. Site-to-site offsets increased again during the latest Miocene and Pliocene, ~5.5 – 3 Ma. From a low in the early Pleistocene (~2.5 Ma) both sites display a near-synchronous increase in CVI that peaked during the middle Pleistocene (~1.2 Ma). A steady decrease in CVI followed during the late Pleistocene at both sites; this trend appears steeper at Site 707 so that mean CVI values diverged again between sites during the past ~1 million years.

### 3.3 Phenotypic evolution at the morphospecies level

Both sites reveal distinct temporal patterns in *Helicosphaera* morphospecies composition (Fig. 5; Fig. S2). In total, 9 morphospecies were observed: *H. carteri, H. hyalina, H. wallichii, H. inversa, H. sellii, H. ampliaperta, H. granulata, H. euphrates* and *H. intermedia*. Of these, three species (*H. carteri*, *H. sellii* and *H. granulata*; Fig. 1) were the most abundant, with common trends in relative abundances over the ~15 Ma interval (Fig. 5a).

*Helicosphaera granulata* was the most dominant species at the South Atlantic site during the middle Miocene (~15-11 Ma) and occasionally within the same period (~14.5 and ~12 Ma) at Site 707. The rest of the interval was dominated by *H. carteri*, with higher contributions of *H. sellii* during the Pliocene and Pleistocene (~4.5-1 Ma). Of the three studied species, only *H. carteri* was present continuously during the past 15 Ma (Fig. 5b; Fig. S1). *Helicosphaera granulata* was present until ~2-3 Ma, whereas *H. sellii* had its first occurrence at ~15 Ma at Site 525 and ~12 Ma at Site 707 and its last occurrence at ~1 Ma at both sites. Interestingly, extinctions of *H. sellii* and *H. granulata* were preceded by a decrease in their mean CVI at both sites (Fig. 5c, d). Similarly, *H. carteri* has experienced a distinct decline in CVI over the past million years and modern *H. carteri* strains have lower mean size and volume than most of their fossil sample counterparts (Fig. 3; Fig. 5b).

Consistently, mean CVI of *H. carteri*, *H. sellii* and *H. granulata* was larger at Site 525 than at Site 707 (*H. carteri*: t = 10.83, df = 2427.2, *p*-value < 0.001; *H. sellii*: t = 7.96, df = 324.91, *p*-value < 0.001; *H. granulata*: t = 15.15, df = 890.57, *p*-value < 0.001). Nevertheless, the long-term species-level patterns, as well as site-to-site offsets, were comparable among the three species (Fig. 5b-d), demonstrating that the genus-level coccolith volume trends and offsets between the sites (Fig. 4a) were not merely caused by relative contributions of differently-sized species, but also by phenotypic evolution at the species-level.

### 3.4 Paleo-ecological records

Stark differences in nannofossil community composition and nannofossil fluxes reflect the contrasting environmental conditions between sites (Fig. S3). Site 525 had consistently higher nannofossil fluxes (average $6.49 \times 10^9$ N m$^{-2}$yr$^{-1}$) than Site 707 (average $2.98 \times 10^9$ N m$^{-2}$yr$^{-1}$) and was dominated by the opportunistic genera *Reticulofenestra* and *Gephyrocapsa* that





bloom in the upper-photic zone, while assemblages at Site 707 were dominated by the species *Florisphaera profunda* that is restricted to the deep-photic zone. Abundances of *Helicosphaera* spp. were relatively low (<4% of assemblage), but it was

consistently present at both sites with average fluxes of $7.57 \times 10^8$ N m$^{-2}$yr$^{-1}$ (Site 525) and $2.30 \times 10^8$ N m$^{-2}$yr$^{-1}$ (Site 707). Other heavily calcified taxa such as *Coccolithus pelagicus* s.l. (Coccolithaceae) had highest abundance (5-15% of assemblage) and fluxes (up to $1.92 \times 10^9$ N m$^{-2}$yr$^{-1}$) at Site 525 prior to ~7 Ma.

*Florisphaera profunda*-based primary productivity estimates (Fig. 4c) and fluxes of upper-photic zone taxa ("total placolith fluxes" in Fig 4d) confirm that significant site-to-site differences in primary production were maintained throughout the past

15 million years. High %*Fp* (up to 75-80%) at Site 707 indicates an oligotrophic system with low to moderate phytoplankton production in the upper photic zone (PP average $131\pm34$ g C m$^{-2}$ yr$^{-1}$) compared to the more productive Site 525 where %*Fp* was consistently <25% and, therefore, PP >200 g C m$^{-2}$ yr$^{-1}$ (average $340\pm65$ g C m$^{-2}$ yr$^{-1}$; Fig 4c; Fig. S3). The highest nannofossil fluxes were recorded during the late Miocene and earliest Pliocene (~9 and 4 Ma), when lowered %*Fp* values also corroborate elevated PP levels at both sites. Despite the highly different community structures and productivity, a similar

transition from larger *Reticulofenestra* species that dominated the middle Miocene community to smaller *Gephyrocapsa* spp. in the late Pliocene and Pleistocene is observed at both sites (Fig. S4).

## 4. Discussion

### 4.1 Coccolith dimensions as a proxy for cell physiology?

Mean coccolith dimensions (CVI, AR or any other parameters tested) in modern *Helicosphaera carteri* strains did not reflect

the significant differences in growth rates, cell size or PIC and POC production rates measured in the same strains (Šupraha et al., 2015). This means that, when analyzed on the cellular and genotype level, coccolith volume cannot be used to indirectly infer cell size, carbon production rates or growth rates in this species. This observation confirms that intracellular coccolith production in *Helicosphaera* is constrained by genetics of biomineralization and defined by the dimensions of the organic template and coccolith vesicle, not by growth and other physiological rates (Henriksen et al., 2004; Young et al., 2004).

Similar to *H. carteri*, mean CVI has no correlation to physiological rates or cell size in single genotypes of the heavily calcified species *Coccolithus pelagicus* and *Coccolithus braarudii* (Gerecht et al., 2014). Nevertheless, a strong positive relationship between coccolith dimensions and cell size of *Coccolithus* spp. is well established based on fossilized coccospheres, representing multiple genotypes and thousands of generations on geological timescales (Gibbs et al., 2018; Henderiks, 2008). Likewise, a positive correlation of coccolith size and cell size in *Coccolithus pelagicus* is evident when different genotypes

from field communities and culture experiments are pooled together (Gibbs et al., 2013).

Apparently conflicting results have been reported for the members of the Noelaerhabdaceae family. For example, Fritz (1999) found that coccolith size does not correlate with physiological rates or cell size within a single genotype of *Emiliania huxleyi*,



while studies by Bolton et al. (2016) and McClelland et al. (2016) found significant correlation between coccolith thickness, aspect ratio and the PIC:POC ratio within Noelaerhabdaceae, based on multiple strains from two closely related taxa (*Emiliania* and *Gephyrocapsa*). These observations support our conclusion that correlations between coccolith morphometry, cell size and physiology are weak or not identifiable within single strains (=single genotypes), and that such relationships can only be established when considered across different genotypes or closely-related species and lineages (Fig. S5).

Across the modern representatives of the *Helicosphaera* genus, coccolith dimensions and cell size are correlated. For example, the smallest coccolith and coccosphere sizes are found in *H. pavimentum* (coccolith: 4-6 µm, coccosphere: 9-16 µm) and in *H. hyalina* (coccolith: 5-8 µm, coccosphere: 11-16 µm) whereas the largest coccolith and coccosphere sizes are found in *H. carteri* (coccolith: 7-12 µm, coccosphere 15-25 µm) and *H. wallichii* (coccolith: 7-12 µm, coccosphere: 19-27 µm) (http://www.mikrotax.org/nannotax3; Geisen et al., 2004). It can therefore be argued with great confidence that these traits followed the same principle in extinct fossil representatives, meaning that *H. sellii* had smaller coccolith and cell size than *H. carteri* while *H. granulata* had the largest cell-size of the three species. Thus, the variations in coccolith dimensions observed in the fossil record can be interpreted as an indicator of changing cell size both at the species- and at the genus level.

## 4.2 Phenotypic evolution during the past 15 Ma: Selection for smaller cells

The long-term phenotypic trends in fossil *Helicosphaera* indicate a selection for smaller cells in the South Atlantic Ocean and the equatorial Indian Ocean over the past 15 Ma. Our dataset supports that this selection for smaller cells occurred on various scales: (I) the (morpho)species level, (II) the community level, as well as (III) the evolutionary level through extinction of larger species and speciation of smaller species.

At the (morpho)species level, this includes selection for smaller-celled phenotypes within each of the three most dominant species (*H. carteri, H. sellii* and *H. granulata*; Fig. 5). Along with the species-level trends, the selection for smaller cell-size is also expressed in the relative contribution of different-sized species and their speciation patterns. This is evident in the gradual decrease in relative abundance and Pliocene extinction of the largest species *H. granulata* whose dominance was gradually replaced by the rise of *H. carteri* and *H. sellii*. Subsequently, the extinction of *H. sellii* in the middle Pleistocene was followed by the rise of an even smaller extant species, *H. hyalina*. The adaptations at the species level, ecological trends on the community level as well as the evolutionary processes (speciation and extinction) on the genus level, all followed the same pattern in favoring smaller-celled genotypes and species within *Helicosphaera*, a trend that is consistent between the two sites.

This selection for smaller morphotypes appears to be a common long-term adaptive response across multiple coccolithophore lineages. For example, the *Calcidiscus leptoporus* species complex (Calcidiscaceae) shows a decrease in the number of radial elements and coccolith diameter since the middle Miocene (Knappertsbusch, 2000; Fig. 6a) and selection for smaller species within the Noelaerhabdaceae family is well documented over the past 15 Ma across multiple deep-sea sites (Bolton et al., 2016; Hannisdal et al., 2012; Suchéras-Marx and Henderiks, 2014; Fig. 6b). The consistent global response across





coccolithophore lineages indicates that the phenotypic evolution of the group is driven by (a) common macroevolutionary
driver(s) that selected for smaller coccolith and thus smaller cell size since the middle Miocene.

### 4.3 What drives the phenotypic evolution of *Helicosphaera* spp.?

The phenotypic expression observed at any point in the fossil record is a mean phenotype of millions of generations of
coccolithophore cells sedimented from the photic zone. As such, it represents the genotypes which at the time were selected
by their environment. It is a result of selection by global and local environmental drivers (e.g. hydrography, temperature,
insolation, nutrients, $CO_2$ levels) as well as evolutionary (extinction, speciation) and ecological processes combined with inter-
and intra-specific interactions such as competition and predation (Falkowski and Oliver, 2007; Finkel et al., 2007). Keeping
in mind this environmental and biological complexity behind the final product – a coccolith with its phenotypic features, what
does the selection for smaller cell size tell us about the possible global and regional drivers of phenotypic evolution in
*Helicosphaera* and in coccolithophores in general?

The past 15 million years of Earth's geological history has included prominent environmental and paleoceanographic changes.
Most importantly, this period saw the transition from the warm and high-$CO_2$ world of the Miocene climatic optimum (17-15
Ma) to the cooler, low-$CO_2$ world of the Pleistocene with its alternating ice ages and interglacial periods (Super et al., 2018;
Zachos et al., 2001; Zhang et al., 2013). The global decrease in $pCO_2$ since the middle Miocene and associated carbon limitation
of photosynthesis and calcification has been suggested as the main global driver of coccolithophore phenotypic evolution
(Bolton et al., 2016). Carbon limitation during the late Miocene may have triggered a physiological adaptive response in
coccolithophores around 7-5 Ma ago in the form of an active allocation of carbon from calcification to photosynthesis, as
inferred from coccolith vital effects (Bolton and Stoll, 2013). A number of studies, spanning beyond the interval addressed
here, have shown that carbon limitation especially affects larger, heavily calcified taxa, due to their low surface-to-volume
(SA:V) ratio and high calcification-demand for carbon (Hannisdal et al., 2012; Henderiks and Pagani, 2007; McClelland et al.,
2016; Pagani et al., 2011). Being among the largest and most calcified species, members of the genus *Helicosphaera* would
therefore be strongly affected by decreasing $CO_2$ levels during the middle to late Miocene, leading to progressive selection for
smaller morphotypes with higher SA:V ratio and lower carbon-demand for calcification. If the adaptive carbon allocation
mechanism observed in Noelaerhabdaceae and Coccolithaceae has also developed in *Helicosphaera* spp., it may account for
the apparent reversal in global phenotypic trends during the late Pliocene and early Pleistocene found in our dataset. Also, it
may indicate that other global environmental drivers and adaptive responses were at play in the most recent time intervals.

While the global decrease in coccolith and cell size across coccolithophore lineages is likely related to decreasing $pCO_2$, the
site-to-site contrasts observed in this study could be related to other environmental drivers which commonly shape
coccolithophore communities such as nutrient availability and temperature. Experiments on *H. carteri* have shown that this
species can strongly reduce its nutrient requirements, arguably as an adaptation to growth in oligotrophic environments
(Šupraha et al., 2015). This comes at a price of slower growth rates, on average larger cell size and slower calcification rates,





but conserved rates of photosynthesis, which is prioritized in respect to calcification under nutrient limitation. Similar short-term response patterns, where limited resources are allocated to maintain photosynthesis at the expense of calcification, were observed in semi-continuous cultures of *C. pelagicus* (Gerecht et al., 2015). On the other hand, *E. huxleyi* is notably insensitive to phosphorus limitation (Gerecht et al., 2017; Oviedo et al., 2014) due to its small size (and SA:V ratio), low nutrient
requirements and remarkable activity of its alkaline phosphatase (Riegman et al., 2000). These experimental observations show that nutrient availability can have a strong effect on coccolithophores and that it likely exerts strong selective pressure in the natural environment, forcing the cells to channel limiting nutrients from calcification to photosynthesis to maintain their division rates or to develop physiological adaptations. Even though physiological low nutrient-adaptation was not reflected in the coccolith phenotype of single strains of *H. carteri* and *C. pelagicus*, it is plausible to expect that prolonged nutrient
limitation would have strong effects on size-selection over longer, evolutionary time-scales (Falkowski and Oliver, 2007). In this sense, lower nutrient availability in the oligotrophic Indian Ocean as opposed to the more productive South Atlantic (inferred from modern data as well as our %*Fp*-based PP estimates and nannofossil community composition) can account for the observed general offset in size between the two sites. In other words, lower nutrient concentrations coupled with carbon limitation have likely channeled the Indian Ocean genotypes towards even smaller cell-size compared to their South Atlantic
counterparts. This process was likely supported and even exacerbated by consistently higher temperatures in the tropics ($27\pm1°C$, long-term average ± 1 s.d.; Herbert et al., 2016) compared to the South Atlantic and higher latitudes (Fig. 2b). Cooler sea surface temperatures in the South Atlantic would have favored both the solubility of $CO_2$ in the ocean and coastal upwelling off SW Africa would have supplied nutrients to the photic zone, thus alleviating the effects of resource limitation and allowing for on average larger cell size at the South Atlantic site compared to the Indian Ocean site.

An important adaptive trait to consider in the context of resource limitation is the heteromorphic, haplo-diploid life-cycle that is prevalent among coccolithophores including *Helicosphaera* (Young et al., 2005). The haploid life-phase is generally significantly smaller and less calcified than the diploid life-phase, and is considered an adaptation to seasonal resource limitation (D'Amario et al., 2017; Šupraha et al., 2016). In addition, both life-cycle phases of *Helicosphaera* are flagellated and have a haptonema, which is a strong indication that they are putative mixotrophs (as demonstrated for *Coccolithus*
*pelagicus*; Houdan et al., 2006) and can relieve nutrient limitation by ingesting bacteria. However, due to their sensitivity to dissolution, holococcoliths of *Helicosphaera* are scarce or near-absent from the fossil record. It is thus difficult to assess how important the life-cycle dynamics of this genus were during its evolutionary history, but given the prevalence of this adaptive strategy in the modern ocean it is safe to assume that it was an important mechanism for overcoming resource limitation also in the past.

**4.5 Biogeochemical implications of phenotypic evolution**

Despite the evolutionary selection for smaller coccolith- and cell size in *Helicosphaera* since the middle Miocene, the trend in AR was less pronounced, indicating relatively consistent biogeochemical output (i.e. similar PIC:POC ratio; Fig. 7) of the



genus. The observed AR trend suggests a slight increase in PIC:POC over time, which could mean that on the cellular level, the adaptive strategy of decreasing coccolith and cell size compensated for resource limitation and resulted in conserved physiological performance. Moreover, the regional trends in size selection were not reflected in regional patterns of AR, which were largely overlapping between both sites throughout the studied interval. Evidently, any global or regional patterns in phenotypic evolution, speciation or adaptation (either in size or physiology) ultimately led to very similar biogeochemical performance among the populations in different regions. A significant divergence in AR between the sites happened in the late Pleistocene, with an increase in AR (and thus PIC:POC) at Site 707. This coincided with an increase in PP estimate as well as coccolith size (Fig. 4), suggesting that locally weaker resource limitation could have led to increasing calcification rates and some degree of regional trends was possible despite overall globally conserved biogeochemical performance.

However, when considered in a wider phylogenetic context and compared with AR patterns observed in the Noelaerhabdaceae (Fig. 6c), the AR trends observed in *Helicosphaera*, though statistically significant, appear biologically meaningless. During the same time interval, the AR and thus the biogeochemical output of Noelaerhabdceae exhibits remarkable variability and a wide range of possible PIC:POC ratios (i.e. degrees of calcification) with an overall decrease between the middle Miocene and Pleistocene. This phenotypic variation is likely related to the overall higher taxonomic and morphological diversity of the Noelaerhabdaceae and it points to a high adaptive potential of this group that ensured its evolutionary success and global dominance during the Neogene, not just within coccolithophore communities (Figs S3 and S4) but in marine phytoplankton in general. Our data show that despite the global trends of size decreases across coccolithophore lineages, phenotypic evolution in different groups can lead to highly different biogeochemical outputs (Fig. 6c). A prime example of this is the comparison between the *Helicosphaera* genus, with its highly conserved morphology, little phenotypic innovation, low abundances and relatively stable biogeochemical performance; and the Noelaerhabdaceae, with the potential for inter-species hybridization (Bendif et al., 2015) and rapid diversification (Bendif et al., 2019), global dominance in marine phytoplankton communities (Suchéras-Marx and Henderiks, 2014; Fig. S3) and a range of possible biogeochemical outputs (Fig. 7; Fig. S5). Clearly, there are various adaptive strategies to the same global drivers among coccolithophore lineages, and a wide range of biogeochemical consequences of their phenotypic evolution.

One possible explanation for different adaptive strategies can be related to the distinction between obligate calcifiers, including *Coccolithus* and *Helicosphaera*, and non-obligate calcifiers such as *Emiliania huxleyi* (Durak et al., 2016; Walker et al., 2018). The phenotypic plasticity in obligate calcifiers could be more restricted when it comes to morphological innovation and biogeochemical performance compared to non-obligate calcifiers, who can regulate their calcification rates (and thus PIC:POC ratio) to zero and overcome carbon limitation or other environmental stressors such as ocean acidification (Gafar et al., 2019). Obligate calcifiers such as *Coccolithus*, *Calcidiscus* and *Helicosphaera*, combined, were relatively more abundant than reticulofenestrids until ~7 Ma when their fluxes decreased significantly at both sites (Fig. S3). This supports that obligate calcifiers have been evolutionary and ecologically less successful compared to non-obligate calcifier lineages during the global



shift to the "icehouse" world of the late Miocene to Pleistocene, each of the groups adapting within the constraints of their phenotype.

## 5. Conclusions

Coccolithophore communities are globally linked on evolutionary time scales through speciation and extinction events. On millennial time scales, environmental changes cause selective pressures which result in common signals of phenotypic
evolution across different coccolithophore lineages, as shown for *Helicosphaera* (Zygodiscales order), *Calcidiscus* (Coccolithales order) and the ancestral lineage of *Emiliania* and *Gephyrocapsa* (Isochrysidales order). However, the biogeochemical output of the group is defined by lineage-specific physiology and adaptive strategies, regionally distinct community compositions and local ecophysiological adaptations. Inferring physiological adaptive strategies from coccolith morphometry proves to be challenging if not impossible when considered within single genotypes. Valid correlations between
morphology and physiology emerge only when more genotypes or closely related taxa are pooled together. Accordingly, short-term physiological responses in e.g. cell size or PIC:POC ratios observed in single-genotype experiments do not necessarily translate into long-term phenotypic evolution observed in the fossil record, which is a result of far more complex and long-term selection with multiple biotic and abiotic drivers. To conclude, the possible scenarios for biogeochemical performance of coccolithophores under ongoing global environmental changes are difficult to predict, as they likely rest on specific adaptive
responses and complexities of different lineages that are beyond simple size-adaptation and are shaped by the adaptive potential of different species and groups, and ultimately by the biodiversity and taxonomic structure of coccolithophore communities.

## Data availability

The data will be made available on PANGAEA after the manuscript is published.

## Acknowledgements

This study was supported by the Royal Swedish Academy of Sciences through a grant from the Alice and Knut Wallenberg foundation (KAW 2009.0287) and funding from the Norwegian Research Council (project 197823/V40) to J.H. We are grateful to Prof. Bente Edvardsen for access to laboratory facilities at the Section for Aquatic Biology and Toxicology (AQUA) at the University of Oslo (UiO) as well as helpful discussions during the PhytoSCALE project. We thank Andrea Gerecht (Centre for Ecological and Evolutionary Synthesis, UiO) for her support in conducting the culture experiments.




**Figures and Figure captions**

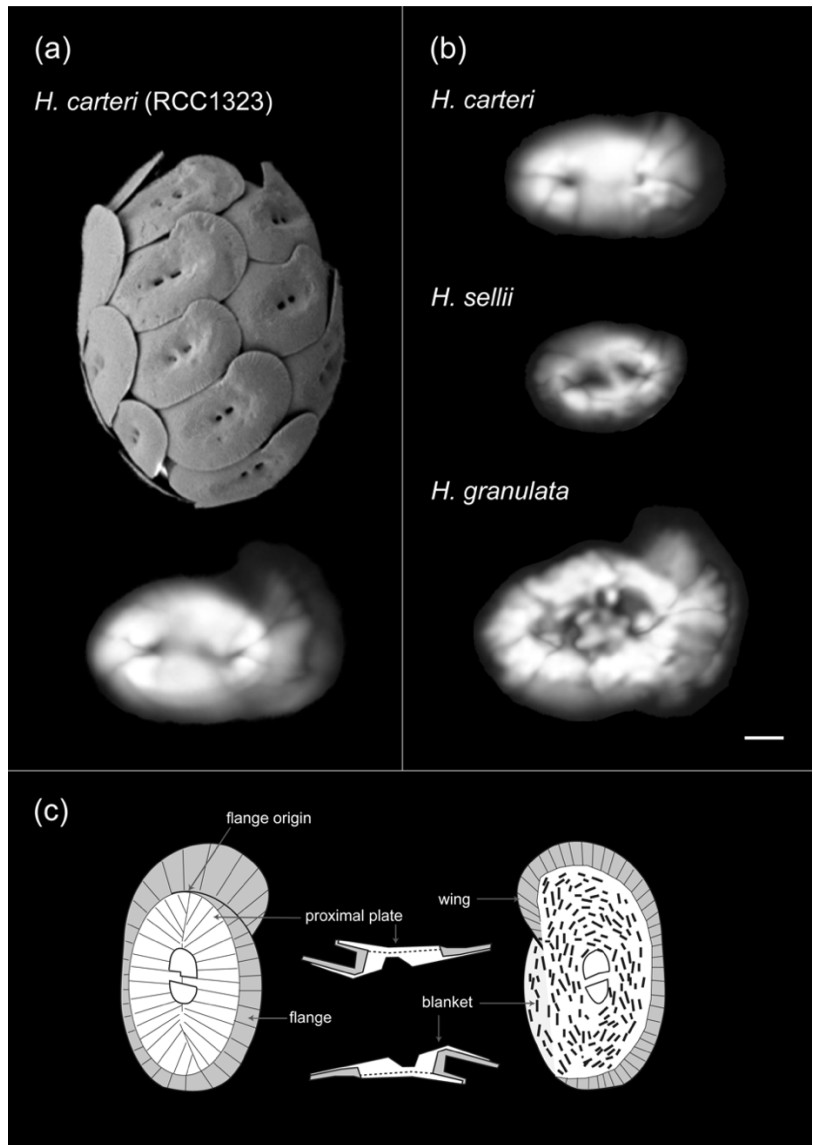


**Figure 1.** *Helicosphaera* coccosphere and coccolith morphology. (a) Scanning electron micrograph of a *Helicosphaera carteri* coccosphere and a polarised light microscope image of an individual coccolith obtained from culture (Atlantic strain RCC 1323) (b) Light microscope images of coccoliths belonging to fossil *Helicosphaera* species analysed in this study: *H. carteri*, *H. granulata* and *H. sellii*. Scale bar=2 μm. (c) Schematic illustration of *Helicosphaera* coccolith morphology and calcite

birefringence under cross-polarised light (after Young et al., 2004). Note the distinction between the birefringent blanket area (white) and the partially birefringent area of the coccolith wing/flange (grey).



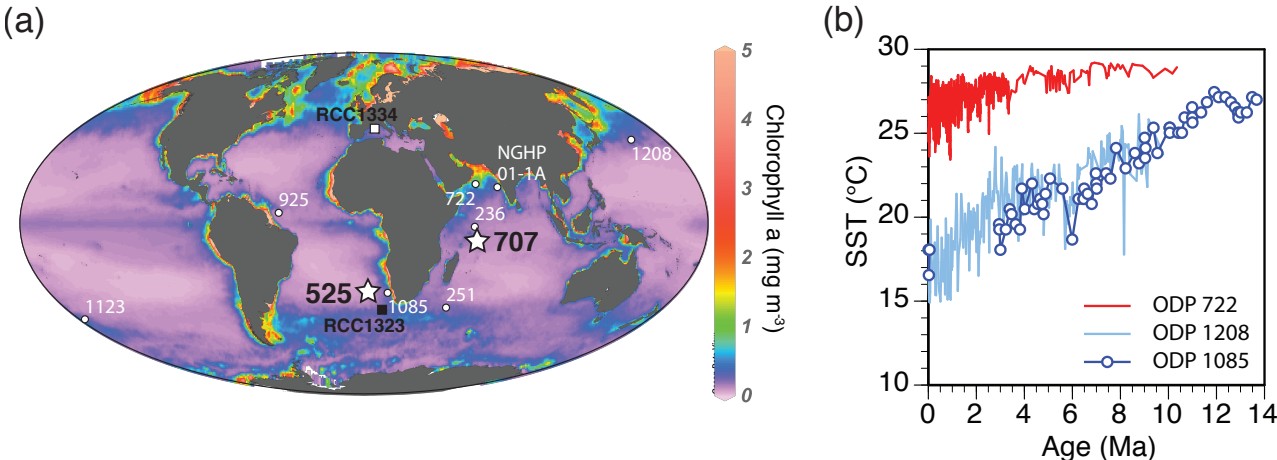

**Figure 2.** (a) Locations of DSDP Site 525 and ODP Site 707 (stars) and other coring sites (open circles) under discussion, as
well as isolation sites of RCC cultures (squares) analysed in Šupraha et al. (2015). Map in Mollweide projection with an
overlay of annual average Chlorophyll *a* concentrations (data obtained from the Ocean Color Data database, NASA,
http://oceancolor.gsfc.nasa.gov/, 2018) was created using the Ocean Data View 4 software (Schlitzer, R., Ocean Data View,
https://odv.awi.de, 2018). (b) Long-term trends in sea surface temperature (SST) at selected temperate and tropical sites (data
from Herbert et al., 2016).



**Figure 3.** Coccolith allometry in modern strains and fossil specimens of *H. carteri*. (a-b) Natural log-transformed (ln) coccolith surface area (μm²) vs. thickness (μm) for all individual fossil specimens at Sites 525 (blue; N=1156) and 707 (red; N=1399) and mean (± 1 s.d.) values for South Atlantic (black) and Mediterranean (open square) strains; (c-d) Coccolith volume index (CVI, ln μm³) vs. aspect ratio (AR, unitless). Here, individual specimens are also shown for the modern strains.



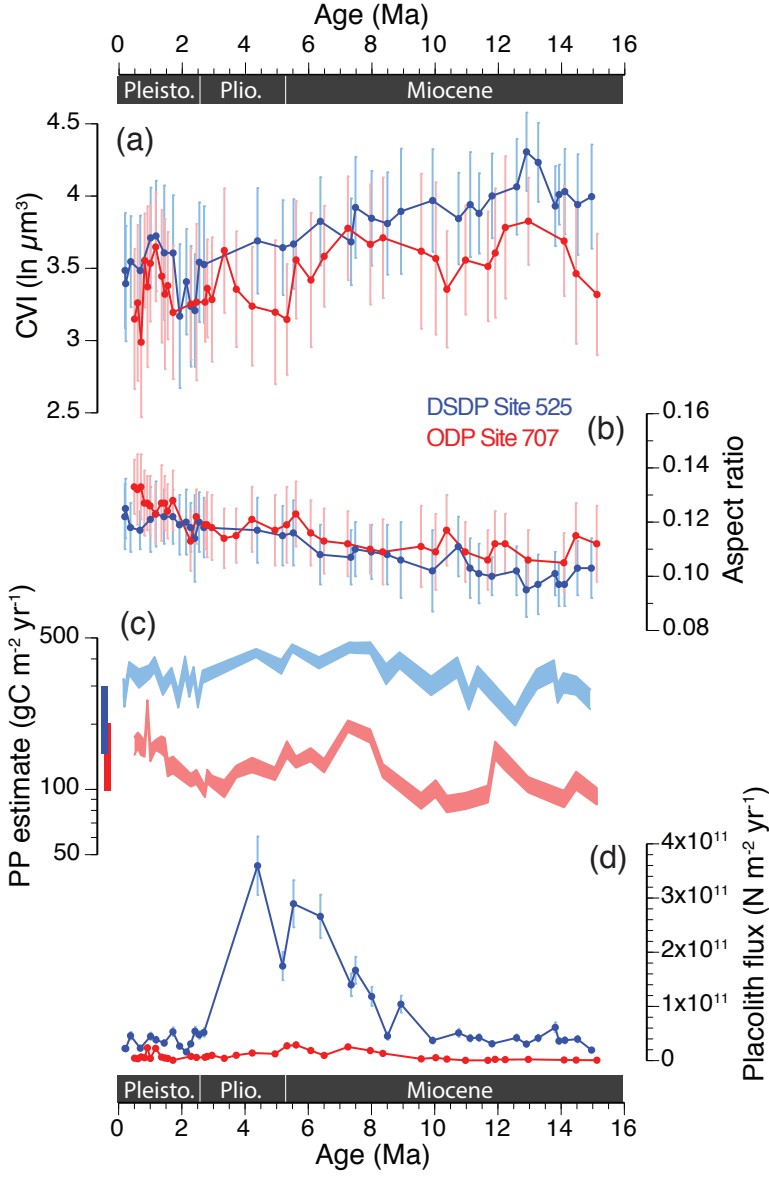

**Figure 4.** Long-term trends in phenotypic evolution of the *Helicosphaera* genus (including all morphospecies) and paleoenvironmental estimates at DSDP Sites 525 (in blue) and ODP Site 707 (in red) over the past 15 Ma. (a) *Helicosphaera* mean coccolith volume index (CVI, ln $\mu m^3$) and (b) coccolith aspect ratio (AR). N≈50 per sample. Error bars show 1 s.d. (c) Primary productivity estimates (PP, gC $m^{-2}$ $yr^{-1}$; 95% confidence intervals) based on the relative abundance of *Florisphaera profunda* (*Fp*%) *cf.* Beaufort et al., (1997). Bars to the left show the range in modern primary production estimates for the South Atlantic (blue) and the Eastern Indian Ocean (red) (Antoine et al., 1996; Beaufort et al., 1997; Fischer et al., 2000). (d) Long-term trends in placolith fluxes (or burial rates, N $m^{-2}$ $yr^{-1}$) of upper-photic zone dwelling species only. Error bars show ±15% reproducibility as determined from repeat analyses (Bordiga et al., 2015).





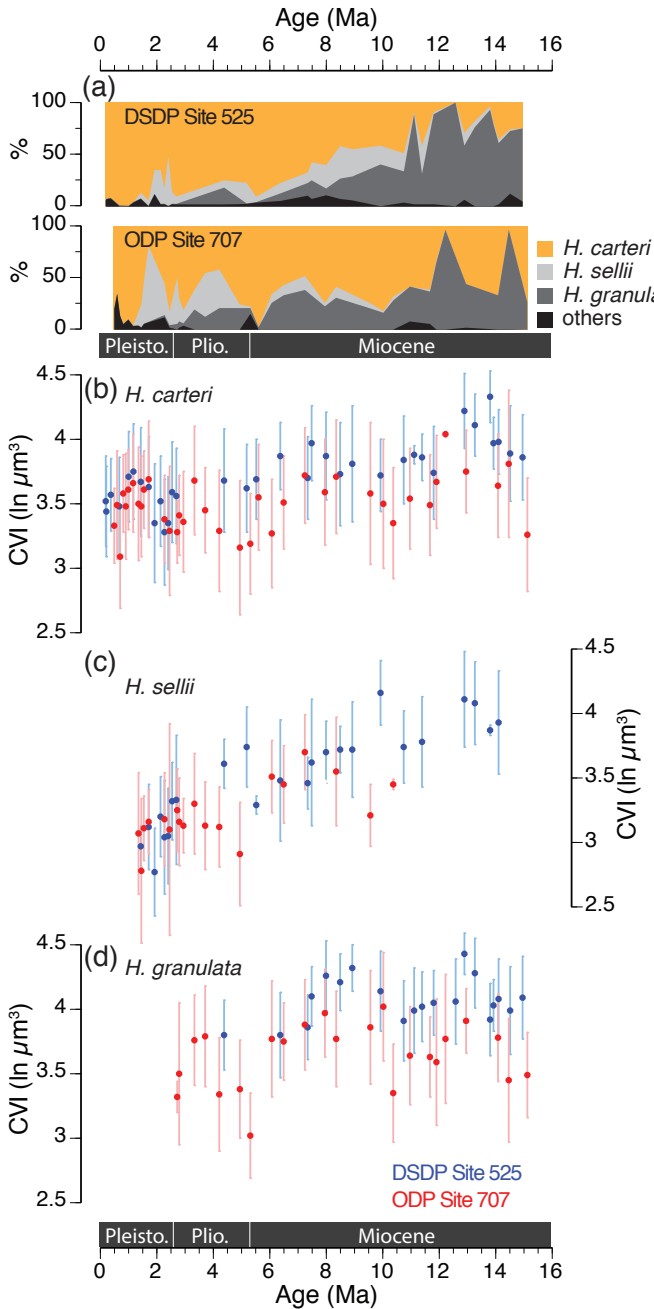

**Figure 5.** Phenotypic evolution of the most abundant *Helicosphaera* fossil morphospecies at DSDP Sites 525 (in blue) and
ODP Site 707 (in red) over the past 15 Ma. (a) Relative proportions (%) of *H. carteri*, *H. sellii*, *H. granulata* and other
*Helicosphaera* morphospecies (closed sum). (b-d) Average CVI (ln μm³) for each of the three most prominent morphospecies
(names indicated in plots). The number of measurements per fossil data point is between 2-50 depending on the relative
abundance of the species in a sample.



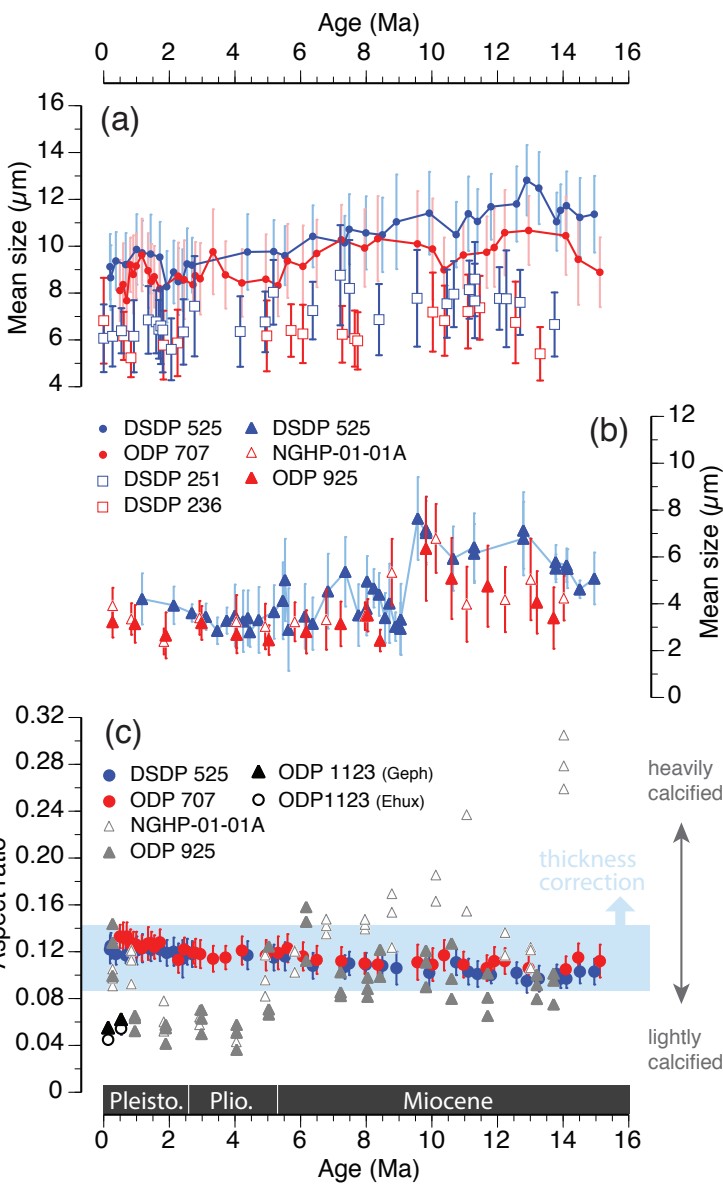


**Figure 6.** Comparison of macroevolutionary phenotypic trends in three prominent coccolithophore lineages at temperate (blue) and tropical (red) sites, over the past 15 Ma. Mean coccolith size (length, µm) of (a) *Helicosphaera* (dots; this study) and *Calcisdicus* (open squares; Knappertsbusch, 2000) and (b) *Reticulofenestra* (triangles) (this study, in blue; Bolton et al., 2016, in red). Error bars are ± 1 s.d. (c) Mean aspect ratio (AR) in fossil specimens of *Helicosphaera* (blue and red dots; as in Fig.

4b) compared to AR estimates in *Reticulofenestra* spp. (grey triangles; grouped within size ranges of 2-3 µm, 3-4 µm, and 4-5 µm, Bolton et al., 2016), *Emiliania* and *Gephyrocapsa* (black symbols; McClelland et al., 2016). The light blue shading encompasses the 1 s.d. variation in *Helicosphaera* over the entire time interval, whereas the long-term average AR would shift upwards if "corrected" for the systematic underestimation of thickness, as discussed in Section 2.3.



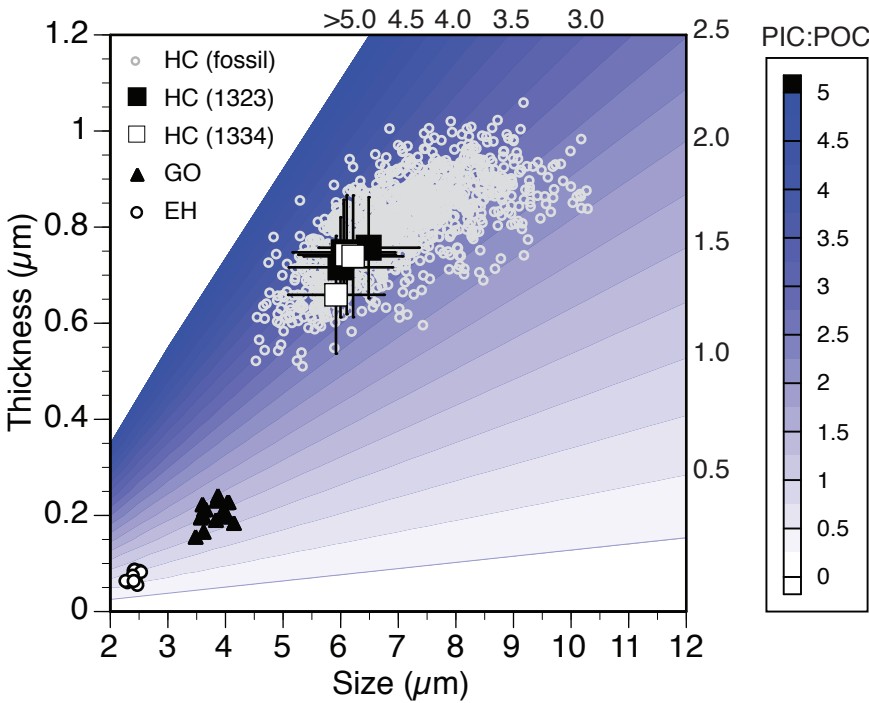

**Figure 7.** *H. carteri* (HC) coccolith dimensions in culture (black squares; mean ± 1 s.d.) and fossil samples (grey circles; individual measurements at DSDP Site 525, N= 1156) overlain onto empirical relationship between coccolith size, coccolith thickness and PIC:POC (contours) based on culture data of *E. huxleyi* (EH) and *G. oceanica* (GO) (McClelland et al., 2016).





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
