# Peer review of "A 15 million-year long record of phenotypic evolution in the heavily calcified coccolithophore *Helicosphaera* and its biogeochemical implications"

_Biogeosciences, 2019_

## Referee Comment (RC1) · Anonymous Referee #1 · 14 Feb 2020

Supraha & Henderiks present a nice new study of the morphological characteristics of modern cultured Helicosphaera carteri coccoliths, and how these parameters relate to each other and to cell physiological parameters (specifically PIC:POC ratios). They then go on to study the geological history and morphological evolution of the Helicosphaera group at two deep-sea coring sites, one in the oligotrophic tropical ocean and one at a higher-productivity mid-latitude site. The study provides an interesting insight into the Neogene evolution of Helicosphera coccolith morphology and discusses potential drivers behind the trends, providing a new perspective when compared to existing work on similar timescales based on the Reticulofenestra lineage. Overall, I think it is a timely and interesting paper that deserves to be published in Biogeosciences following some minor revisions. Specific comments are detailed below.

One general comment relates to the reported abundances of Florisphaera profunda. According to PP estimates in Fig 4, F. profunda coccoliths are present at quite stable and high levels (30% to >60% at Site 707) throughout the Miocene and Pliocene, back to 15 Ma at both sites. This is quite surprising. In my personal experience doing (low-latitude) biostratigraphy I have only observed this species as far back as the late Miocene (∼7-8 Ma) and abundances by then were pretty low. Looking at the Nannotax range chart and other literature sources, these suggest an earliest first common occurrence of F. profunda around 8-10 Ma. So your results are really intriguing to me and I would like to know more! Are these Miocene F. profunda morphologically similar to the Plio-Pleistocene forms? Are you certain that you really have ∼60% (equivalent to a PP of ∼120 gC m-2 yr-1 using the Beaufort 1997 equation) relative abundances of this species in the middle and late Miocene at your Indian Ocean site? Another point related to the difference between the two sites: given the different temperature histories, I would suggest that this also contributes to the lower % F. profunda at site 525 (not just differences in productivity regime) – see this recent compilation paper: https://www.sciencedirect.com/science/article/abs/pii/S0277379118306139.

Comment on figures: I prefer the way you have displayed data in Figure S1 to the main figures (mean and 1 stdev). Maybe instead of the error bars, you could plot the original data behind the means? Or maybe use box and whisker plots or joy plots in the main figures? I just think that the way the data are currently displayed in the main paper does not do justice to all the measurements you did, or accurately show the range/distribution within each sample population. At the moment it looks like your data points have large "errors", when in fact it is just that within-population variability is high. Figure s5 seems important to me, and maybe warrants promotion to the main text.

Abstract: I am not sure I fully understand the term "biogeochemical performance" – is

this one that you are defining here for the first time or is this a commonly-used term in biology? You say that the fact that coccolithophores have a wide range of sizes and degrees of calcification implies that they have high "biogeochemical performance" – does this mean that they are more successful than other species that have smaller ranges of size etc? to me it just implies that they are highly adaptable so can thrive in different environmental conditions. I found this first sentence of the abstract a bit confusing.

Line 17-20: I suggest reversing these sentences so that you talk about the modern species first, then the fossil work.

Line 19: "which displays eco-physiological adaptations in modern strains" – reference missing

Line 21: state which physiological traits are you talking about – or use this term in line 2 (i.e. a wide range in physiological traits such as cell size, degree of calcification [. . ..]). Same with "physiological rates" – be specific what you are talking about.

Line 26: mean coccolith or cell size?

Line 38: Explain more clearly via what mechanisms coccolithophores "are still alleviating the negative effects of rising [. . .] CO2 levels" Is this statement consistent with the following sentence?

Line 54: these 2 references don't really represent all of the temperature and pCO2 data out there.

57: There are also older references that show this size trend that should be included. Also. Bolton et al., 2016 does not reconstruct calcification rates (maybe states? degree of calcification?)

58: Middle Miocene to Pleistocene

89: I assume you mean not nutrient limited

100: "Age models are based on calcareous nannoplankton biostratigraphy" – add references? Or is this your work?

105: reverse transparent and filters.

126: I would put this sentence at the end of the paragraph, once you have discussed all the complications, so the reader is left with the point that relative changes are still accurate. Also, perhaps important to note, did you see any temporal changes in the amount of coloured parts (thicker than 1.55um) on Helicosphaera coccoliths?

158: modern cultured?

174: does local mean site-specific?

194: Not sure what "slightly negative allometry" means. It the cited figures thickness increases with area.

195: Sentences not clear. Are "phenotype dimensions" (of what?) and "mean coccolith traits" the same thing?

201: spelling error. Also replace specimens with coccoliths.

303: older references missing (ex. Young 1990)

Sentence line 327-329: I don't follow this statement. . .

395: evolutionarily

Conclusion: the statement on millennial timescales seems to come out of the blue, I thought you had mainly discussed multi-million-year timescales? Maybe it would clarify if you added the references. I also think that your conclusions are quite general and vague, and could relate more to your finding in this study.

---

## Referee Comment (RC2) · Anonymous Referee #2 · 18 Mar 2020

This morphometric study of the coccolithophore Helicosphaera is extremely interesting. It shows that different coccolithophore lineage adapted differently to the oceanographic changes that occurred in the Late Neogene: The morphological adaptation of Helicosphaera is different from that of Reticulofenestra and Gephyrocapsa. The first group modifies the size of the coccoliths but not their aspect ratio, whereas the second modifies both. Knowing that the aspect ratio is, in coccolithophores, a way to adapt their physiology to environment, this founding is important because it shows that different adaptative strategies are at play in this phytoplanktonic group. The paper is very

well written, the data are abundant and of high quality. The figures are well designed. It is rare that I have to review a manuscript of that quality and with very little to say expect trying to replicate what is already written. My only surprise is to see a record of the percentage of Florisphaera profonda covering the last 15 Ma. In my experience F. profunda first evolved around 10 Ma. So what was counted between 15 and 10 Ma ? Can we show picture of a specimen ? I understand that this comment is not relevant to the main discussion of this manuscript. I congratulate the authors on their work because I have to stop wondering what constructive criticisms I could formulate.
* * *

---

## Author Comment (AC1) · 8 Apr 2020

We are thankful to the two anonymous referees for their thorough reviews and constructive comments on the manuscript. Our point-by-point response is written below:

Supraha & Henderiks present a nice new study of the morphological characteristics of modern cultured *Helicosphaera carteri* coccoliths, and how these parameters relate to each other and to cell physiological parameters (specifically PIC:POC ratios). They then go on to study the geological history and morphological evolution of the *Helicosphaera* group at two deep-sea coring sites, one in the oligotrophic tropical ocean and one at a higher-productivity mid-latitude site. The study provides an interesting insight into the Neogene evolution of *Helicosphera* coccolith morphology and discusses potential drivers behind the trends, providing a new perspective when compared to existing work on similar timescales based on the *Reticulofenestra* lineage. Overall, I think it is a timely and interesting paper that deserves to be published in Biogeosciences following some minor revisions. Specific comments are detailed below.

One general comment relates to the reported abundances of *Florisphaera profunda*. According to PP estimates in Fig 4, F. profunda coccoliths are present at quite stable and high levels (30% to >60% at Site 707) throughout the Miocene and Pliocene, back to 15 Ma at both sites. This is quite surprising. In my personal experience doing (low-latitude) biostratigraphy I have only observed this species as far back as the late Miocene (_7-8 Ma) and abundances by then were pretty low. Looking at the Nannotax range chart and other literature sources, these suggest an earliest first common occurrence of F. profunda around 8-10 Ma. So your results are really intriguing to me and I would like to know more! Are these Miocene F. profunda morphologically similar to the Plio-Pleistocene forms? Are you certain that you really have _60% (equivalent to a PP of _120 gC m-2 yr-1 using the Beaufort 1997 equation) relative abundances of this species in the middle and late Miocene at your Indian Ocean site?

**Response:** Both referees questioned the presence of *F. profunda* in the samples older than the common first occurrence date of this species (around 8-10 Ma; nannofossil zone NN10). To address this issue, we have thoroughly re-examined samples at both sites, paying particular attention to the time interval between 6-12 Ma (nannofossil stratigraphic zones NN11 and NN10). We aimed at identifying the most probable first occurrence of *F. profunda* and documenting any *Florisphaera*-like particles that may have been misidentified as *F. profunda* during the initial counting procedure. During the re-examination, we collected images of typical *F. profunda* morphotypes, and various *F. profunda*-like particles. They are now shown in Fig. 1 and Fig. 2 of the response document.

While counting, we applied the commonly used criteria for identifying *F. profunda* under the polarized-light microscope: (I) the characteristic shape and size of nannoliths (which proved to be highly consistent throughout the interval) (II) low birefringent, greyish appearance under the polarized light, and (III) the "single calcite crystal" morphology, which renders characteristic extinction patterns when parallel to the crossed nicols. Fig.1a/b and Fig.2a show typical specimens of *F. profunda* found in the most recent (Pliocene and Pleistocene) samples. In the late Miocene samples, *F. profunda* specimens become smaller and thicker, likely indicating different phenotypes and possible overgrowth (Fig. 1c, Fig. 2b). At Site 525, *F. profunda* morphotypes become very rare during NN11 (around 7-8 Ma), suggesting that this interval likely represents the first occurrence of the species at this site. At Site 707, there is no such pronounced decrease in abundance of *F. profunda* with time. Still, the relative abundance of the typical Pliocene-Pleistocene morphotype reaches its minimum (21%) at similar time-point as at Site 525 (Around 7-8 Ma during NN11).

**Figure 1.** Morphology of *Florisphaera profunda* and *Florisphaera*-like particles at Site 525. Scale bar = 5μm.

[Figure]

| 0.36 Ma NN20 | 2.26 Ma NN18 | 7.99 Ma NN11 | 8.49 Ma NN10 | 12.57 Ma NN6 |

**Figure 2.** Morphology of *Florisphaera profunda* and *Florisphaera*-like particles at Site 707. Scale bar = 5μm.

[Figure]

| 0.59 Ma NN19 | 6.49 Ma NN11 | 8.35 Ma NN11 | 9.56 Ma NN10 | 10.02 Ma NN10 |

When first analyzing our samples from younger to older material and using the consistent identification criteria, we encountered a range of *Florisphaera*-like particles well below NN11, and we initially decided to include them into our *F. profunda* record. Some examples of these specimens, most similar to typical *F. profunda*, can be seen in Fig.1 d/e and Fig. 2 c/d/e. While their shape and appearance under polarized light are very similar to *F. profunda*, they are usually thicker and exhibit more variation than typical *F. profunda*. Notably, they often have a more elongated appearance, with a pointed distal end of the "nannolith" (e.g. Fig. 2e). Interestingly, after 8.5 Ma, these *Florisphaera*-like particles increase in abundance at both sites, though they are much more abundant at Site 707. While we acknowledge that they likely do not represent *F. profunda* in the strict sense, they overlap with "true" *F. profunda,* especially at its lowest occurrence in NN10 (at site 707).

Determining the identity of the *Florisphaera*-like particles is beyond the scope of this study. Therefore, in our revisions, we decided to show the data for which we can unambiguously claim that it represents *F. profunda*. Our *F. profunda* record will thus start at around 7-8 Ma (NN11) in the revised manuscript (Figure 4c). Nevertheless, the identity of the mysterious *Florisphaera*-like particles is indeed intriguing and remains to be addressed in future taxonomic studies. In any case, as also pointed out by Referee #2, the shorter record of

*F. profunda* in the revised manuscript does not affect our argument about the contrasting primary production levels between the sites. These contrasts are still reflected in the revised %Fp (Fig. 4c), placolith fluxes (Fig. 4d) as well as the relative abundance of typical oligotrophic genera *Sphenolithus* spp. and *Discoaster* spp. (Supplementary Fig. S3).

Another point related to the difference between the two sites: given the different temperature histories, I would suggest that this also contributes to the lower % F. profunda at site 525 (not just differences in productivity regime) – see this recent compilation paper: https://www.sciencedirect.com/science/article/abs/pii/S0277379118306139.

**Response:** We refer to this manuscript in the revised Methods section, and take into account the effect of temperature at higher-latitude Site 525, when using the %Fp to infer the primary production.

Comment on figures: I prefer the way you have displayed data in Figure S1 to the main figures (mean and 1 stdev). Maybe instead of the error bars, you could plot the original data behind the means? Or maybe use box and whisker plots or joy plots in the main figures? I just think that the way the data are currently displayed in the main paper does not do justice to all the measurements you did, or accurately show the range/distribution within each sample population. At the moment it looks like your data points have large "errors", when in fact it is just that within-population variability is high.

**Response:** The revised, main-text Figure 3a, now shows the original raw data (as in Figure S1) along with the mean trends and without the error bars. In Figure 4 in the main text (showing species-level trends), we decided to keep the original formatting. The raw, species-level data is shown in the supplementary figure S2.

Figure S5 seems important to me, and maybe warrants promotion to the main text.

**Response:** Figure S5 is now included in the main text.

Abstract: I am not sure I fully understand the term "biogeochemical performance" – is this one that you are defining here for the first time or is this a commonly-used term in biology?

**Response:** The term "biogeochemical performance" is now replaced with the commonly-used term "biogeochemical impact" throughout the text.

You say that the fact that coccolithophores have a wide range of sizes and degrees of calcification implies that they have high "biogeochemical performance" – does this mean that they are more successful than other species that have smaller ranges of size etc? to me it just implies that they are highly adaptable so can thrive in different environmental conditions. I found this first sentence of the abstract a bit confusing.

**Response:** The term "biogeochemical performance" in our context is a synonym of the term "biogeochemical impact" i.e. the cumulative effect that coccolithophores have on the biogeochemical carbon cycle. This effect is defined by the abundance of different coccolithophore species and their biogeochemically relevant traits (e.g. PIC and POC production rates). The first sentence of the abstract is now clarified to better convey our message.

Line 17-20: I suggest reversing these sentences so that you talk about the modern species first, then the fossil work.

**Response:** The primary focus of this manuscript is on the long-term phenotypic evolution of *Helicosphaera*. Therefore, we decided to keep these sentences as they are.

Line 19: "which displays eco-physiological adaptations in modern strains" – reference missing

**Response:** It is a common practice not to use references in the abstract. This statement is repeated and supported with references in the Introduction section.

Line 21: state which physiological traits are you talking about – or use this term in line 2 (i.e. a wide range in physiological traits such as cell size, degree of calcification [: : :.] ). Same with "physiological rates" – be specific what you are talking about.

**Response:** Following the reviewer´s recommendations, defined the important physiological traits earlier in the abstract.

Line 26: mean coccolith or cell size?

**Response:** This sentence is now changed to: "However, despite a significant decrease in mean coccolith and cell size…".

Line 38: Explain more clearly via what mechanisms coccolithophores "are still alleviating the negative effects of rising [: : :] CO2 levels" Is this statement consistent with the following sentence?

**Response:** We have clarified these statements by distinguishing between the net biogeochemical impact on ocean chemistry (calcification, photosynthesis, carbon burial), alleviating effects on global climate (by ballasting of organic carbon export production) and cellular biogeochemical output defined by the PIC/POC ratio. The revised sentence now reads "…and they are still alleviating the negative effects of rising atmospheric and oceanic $CO_2$ levels by the ballasting of organic carbon export production (Ridgwell and Zeebe, 2005). Their cellular biogeochemical output, which is commonly summarized as a balance of inorganic (PIC) and organic (POC) carbon production rates (*i.e.* PIC:POC ratio)…".

Line 54: these 2 references don't really represent all of the temperature and pCO2 data out there.

**Response:** We have added the following references: Herbert et al 2016, Cramer et al 2009, and Zhang et al. 2013, to support our statement about the Miocene to Pleistocene climate transition (ocean temperature and $CO_2$).

57: There are also older references that show this size trend that should be included. Also. Bolton et al., 2016 does not reconstruct calcification rates (maybe states? degree of calcification?)

**Response:** We have included additional references documenting coccolith size-trends since the Middle Miocene and clarified the reference to Bolton et al. 2016. The revised sentence now reads: "…as also evidenced by a macroevolutionary decrease in cell size (Young 1990; Knappertsbusch et al 2000; Suchéras-Marx and Henderiks, 2014; Imai et al 2015; Bolton et al. 2016) and degree of calcification (Bolton et al., 2016).".

58: Middle Miocene to Pleistocene

**Response:** Corrected. The sentence now reads:"…which dominated coccolithophore communities during the Middle Miocene to Pleistocene".

89: I assume you mean not nutrient limited

**Response:** The corrected sentence reads: "All phenotypic and physiological data used in the present study were obtained from the exponentially growing, non-nutrient limited control experiments only.".

100: "Age models are based on calcareous nannoplankton biostratigraphy" – add references? Or is this your work?

**Response:** A reference to this study, Suchéras-Marx and Henderiks (2014) and to the original shipboard data were added. The revised sentence reads: "Age models are based on calcareous nannoplankton biostratigraphy (this study; Suchéras-Marx and Henderiks, 2014; Backman et al., 1988; Shackleton et al. 1984), updated to the most recent geological time scale (Gradstein et al., 2012) (Table S1).".

105: reverse transparent and filters.

**Response:** We are thankful to the reviewer for spotting this. The corrected sentence reads: "Filters were dried at 60°C and mounted with Canada Balsam (Merck, USA) on microscope slides, rendering filters transparent under polarized light.".

126: I would put this sentence at the end of the paragraph, once you have discussed all the complications, so the reader is left with the point that relative changes are still accurate.

**Response:** A statement conveying a similar message to the first sentence is repeated at the end of the paragraph, stating that the "Statistically significant changes in CVI are meaningful indicators, if not a muted expression, of the true phenotypic changes in both modern and fossil *Helicosphaera* assemblages.". We hope that this is a clear pointer for readers to follow the rest of the text.

Also, perhaps important to note, did you see any temporal changes in the amount of coloured parts (thicker than 1.55um) on Helicosphaera coccoliths?

**Response:** Regarding the temporal changes in the amount of coloured, thicker parts, they increase with age, following the general increase in coccolith thickness and increased relative contribution of thicker morphospecies such as *H. granulata*. Again, this increase in coloured parts suggests that the signal observed in our study is indeed "muted" i.e. the "real" thickness in older samples is likely underestimated. We also address this issue in original Figure 6c by stating that "the long-term average AR would shift upwards if "corrected" for the systematic underestimation of thickness, as discussed in Section 2.3.".

158: modern cultured?

**Response:** Corrected. The sentence reads: "…with PIC:POC ratios measured in modern cultured strains of *Emiliania huxleyi* and *Gephyrocapsa oceanica*, deriving the following power-law relationship:"

174: does local mean site-specific?

**Response:** For better understanding, "local" was changed to "site-specific" in this sentence: "Fluxes of *Helicosphaera* spp. and other taxa (in N m$_{-2}$ yr$_{-1}$) were calculated from the absolute abundances and the site-specific mass accumulation rates.

194: Not sure what "slightly negative allometry" means. It the cited figures thickness increases with area.

**Response:** Yes, that is true; thickness does increase with area. An isometric scaling between area (µm2) and thickness (µm) would give a square-root (exponent 0.5) relationship between the two parameters. However, the logged data shown in Figure 3, render slopes (and thus exponents) of 0.386 and 0.397, which is lower than 0.5. That is why we refer to it as "slightly negative" allometry (non isometric). In our revised manuscript, we will rephrase "slightly negative allometry" to more general terms.

195: Sentences not clear. Are "phenotype dimensions" (of what?) and "mean coccolith traits" the same thing?

**Response:** For clarity, we changed "phenotype dimensions" and "mean coccolith traits" to "coccolith dimensions".

201: spelling error. Also replace specimens with coccoliths.

**Response:** Corrected.

303: older references missing (ex. Young 1990)

**Response:** As with the previous, related comment, we expanded the reference list to include other, older publications (Young 1990; Hannisdal et al., 2012; Suchéras-Marx and Henderiks, 2014; Imai et al., 2015; Bolton et al. 2016).

The Sentence line 327-329: I don't follow this statement:

**Response:** We agree with the reviewer that the sentence does not follow logically from previous statements. This paragraph will be re-written in the revised manuscript to clarity the argument.

In this paragraph, we discuss the apparent reversal of the long-term size-decrease in *Helicosphaera* coccolith size during the Pleistocene. A similar reversal was observed in the reticulofenestrid record (Bolton and Stoll 2016). In that paper, the authors argue that some other factors (i.e. ocean alkalinity) may have caused the increase in coccolithophore calcification and coccolith size, despite decreasing pCO$_2$.

395: evolutionarily

**Response:** Corrected.

Conclusion: the statement on millennial timescales seems to come out of the blue, I thought you had mainly discussed multi-million-year timescales? Maybe it would clarify if you added the references. I also think that your conclusions are quite general and vague, and could relate more to your finding in this study.

**Response:** The "millennial timescales" statement is now changed to "multi-million-year timescales", which are indeed applicable to this manuscript. In the Conclusions section, we aimed at placing our findings in a more general context rather than repeat the main results of the paper. In the revised manuscript, we will re-write the conclusions to focus on *Helicosphaera*, while aiming to still keep the more general perspective.

**Anonymous Referee #2**

This morphometric study of the coccolithophore *Helicosphaera* is extremely interesting. It shows that different coccolithophore lineage adapted differently to the oceanographic changes that occurred in the Late Neogene: The morphological adaptation of *Helicosphaera* is different from that of *Reticulofenestra* and *Gephyrocapsa*. The first group modifies the size of the coccoliths but not their aspect ratio, whereas the second modifies both. Knowing that the aspect ratio is, in coccolithophores, a way to adapt their physiology to environment, this founding is important because it shows that different adaptative strategies are at play in this phytoplanktonic group. The paper is very well written, the data are abundant and of high quality. The figures are well designed. It is rare that I have to review a manuscript of that quality and with very little to say expect trying to replicate what is already written. My only surprise is to see a record of the percentage of *Florisphaera profonda* covering the last 15 Ma. In my experience *F. profunda* first evolved around 10 Ma. So what was counted between 15 and 10 Ma ? Can we show picture of a specimen ? I understand that this comment is not relevant to the main discussion of this manuscript. I congratulate the authors on their work because I have to stop wondering what constructive criticisms I could formulate.

**Response:** We are thankful to Anonymous Referee #2 for the very positive assessment of our manuscript. The issue of *F. profunda* is addressed in detail above, in the response to Referee #1.